# Targeted suppression of mTORC2 reduces seizures across models of epilepsy

James Okoh [1,2,9,11], Jacqunae Mays[1,2,11], Alexandre Bacq [3], Juan A. Oses-Prieto [4], Stefka Tyanova[5], Chien-Ju Chen[1,2,10], Khalel Imanbeyev[1,2], Marion Doladilhe[3], Hongyi Zhou[1,2,9], Paymaan Jafar-Nejad[6], Alma Burlingame [4], Jeffrey Noebels[1,7,8], Stephanie Baulac [3] & Mauro Costa-Mattioli [1,2,9] ✉

Epilepsy is a neurological disorder that poses a major threat to public health. Hyperactivation of mTOR complex 1 (mTORC1) is believed to lead to abnormal network rhythmicity associated with epilepsy, and its inhibition is proposed to provide some therapeutic benefit. However, mTOR complex 2 (mTORC2) is also activated in the epileptic brain, and little is known about its role in seizures. Here we discover that genetic deletion of mTORC2 from forebrain neurons is protective against kainic acid-induced behavioral and EEG seizures. Furthermore, inhibition of mTORC2 with a specific antisense oligonucleotide robustly suppresses seizures in several pharmacological and genetic mouse models of epilepsy. Finally, we identify a target of mTORC2, Nav1.2, which has been implicated in epilepsy and neuronal excitability. Our findings, which are generalizable to several models of human seizures, raise the possibility that inhibition of mTORC2 may serve as a broader therapeutic strategy against epilepsy.

Epilepsy, one of the most common neurological disorders, globally affects over 50 million people[1]. As a result, it poses a major threat to public health and is responsible for an enormous economic and social burden (the global cost of epilepsy is estimated to be >$119 billion annually)[2]. The hallmark of epilepsy is the presence of spontaneous recurrent seizures resulting from hypersynchronous discharge of neurons in the brain. More recently, the definition of epilepsy has expanded to include the adverse neurobiological, cognitive, psychological and social consequences accompanying the seizures observed in patients[3,4]. While some anti-seizure medications have proven beneficial for the treatment of seizures, they often carry serious adverse effects[5]. Importantly, over one-third of the patients suffering from epilepsy do not respond to currently available treatments[6]. Thus, the

identification of new antiepileptic targets and novel antiseizure strategies is of crucial importance.

Over the last few years, whole-genome sequencing studies have identified specific genes and protein networks whose dysfunction has been associated with epilepsy[7,8]. One such pathway is the mammalian target of rapamycin (mTOR, now re-defined as the mechanistic target of rapamycin). mTOR, a serine/threonine protein kinase, constitutes the catalytic subunit of two distinct complexes known as mTOR complex 1 (mTORC1) and 2 (mTORC2)[9,10]. These complexes are distinguished by their molecular constituents and sensitivity to rapamycin. mTORC1 contains the defining component Raptor (regulatory-associated protein of mTOR) and is responsive to acute rapamycin inhibition whereas mTORC2, which was more recently discovered,

[1]Department of Neuroscience, Baylor College of Medicine, Houston, TX, USA. [2]Memory and Brain Research Center, Baylor College of Medicine, Houston, TX, USA. [3]Institut du Cerveau-Paris Brain Institute-ICM, Sorbonne Université, Inserm, CNRS, Hôpital de la Pitié Salpêtrière, F-75013 Paris, France. [4]Departments of Chemistry and Pharmaceutical Chemistry, University of California San Fransisco, San Fransisco, CA, USA. [5]Altos Labs Inc, Bay Area Institute, Redwood City, CA, USA. [6]Ionis Pharmaceuticals, Carlsbad, CA, USA. [7]Department of Molecular & Human Genetics, Baylor College of Medicine, Houston, TX, USA. [8]Department of Neurology, Baylor College of Medicine, Houston, TX, USA. [9]Present address: Altos Labs Inc, Bay Area Institute, Redwood City, CA, USA. [10]Present address: Novartis Inc, Boston, MA, USA. [11]These authors contributed equally: James Okoh, Jacqunae Mays. ✉e-mail: mcostamattioli@altoslabs.com

contains Rictor (rapamycin-insensitive companion of mTOR) and Sin1 (mitogen-activated protein kinase associated protein 1), and is insensitive to acute rapamycin treatment[10].

Loss-of-function mutations in up-stream negative regulators of mTOR, activating mutations in positive regulators of mTOR, and gain-of-function mutations in mTOR itself are associated with epilepsy and cortical dysplasia[11,12]. These "mTORopathies" share two main features: (a) hyperactivation of mTORC1 and (b) drug-resistant epilepsy. In addition to intrinsic mTORopathies of monogenic origin, increased mTORC1 activity has also been reported in the brains of individuals and animal models of temporal lobe epilepsy (TLE)[13–16]. Importantly, chronic treatment with the drug rapamycin (or related compounds known as rapalogs) attenuates the seizure phenotype in genetic models of mTORopathies and models of TLE[12,14–18]. Consequently, the current view is that hyperactivation of mTORC1 leads to increased neuronal excitability and seizures, and its inhibition is a likely beneficial therapy for epilepsy[12,19]. However, increased mTORC2 activity has also recently been reported in individuals with TLE[13] and in those carrying loss-of-function variants in *PTEN*[20], an upstream negative regulator of mTOR. Moreover, chronic rapamycin treatment, which reverses the seizure phenotype in several mTORopathy models[12], also inhibits mTORC2[14,21,22]. Finally, genetic inhibition of mTORC2 reverses the seizure phenotype and behavioral abnormalities in mice lacking *Pten* in forebrain neurons[23]. Thus, it is important to define the contributions of mTORC2 to the seizure phenotype associated with epilepsy. Here, we used overlapping and orthogonal approaches, both scientific and technical, to study the role of mTORC2 in seizures across different domains and levels of analyses. We found that both genetic and therapeutic inhibition (ASO-mediated) of mTORC2 suppresses seizures in multiple models of epilepsy. In addition, we identified Nav1.2 as a potential mechanistic target by which inhibition of mTORC2 reduces seizures.

## Results

### Persistent activation of mTORC2, but not mTORC1, by chemoconvulsant agent kainic acid

To examine the roles of the mTOR complexes in epilepsy, we employed kainic acid (KA), a well-studied model of pharmacoresistant hippocampal epileptogenesis[24,25]. First, we sought to determine whether mTOR complexes exhibit different kinetics of activation upon KA injection. Briefly, mice were injected intraperitoneally (i.p.) with KA (25 mg/kg) and the activity of mTORC1 and mTORC2 was examined at different times post-injection in the cortex and hippocampus. We found that KA activates mTORC1-mediated phosphorylation of ribosomal protein S6 (S6) at Ser240/244 (a well-established readout of mTORC1 activity[9,10]) for up to 24 hours in both the cortex (Fig. 1a, b) and hippocampus (Supplementary Fig. 1a). In contrast, the same treatment activated mTORC2-mediated phosphorylation of Akt at Ser473 (a well-established readout of mTORC2 activity[9,10]) for up to 2 weeks in the cortex (Figs. 1a, 1c) and hippocampus (Supplementary Fig. 1b). Given that KA activated mTORC2 more persistently than mTORC1, we hypothesized that sustained mTORC2 activity may drive KA-induced epileptogenesis.

### Genetic inhibition of mTORC2 suppresses acute KA-induced seizures

To test this hypothesis, we selectively silenced mTORC2 in forebrain neurons using molecular genetics. Briefly, we conditionally deleted *Rictor* (encoding Rictor, a defining component of mTORC2) from the murine forebrain by crossing mice expressing *Cre* recombinase under the control of the α subunit of the calcium/calmodulin-dependent protein kinase II (CamKIIα) promoter with *Rictor*[loxP/loxP] mice to generate *Rictor* forebrain-specific knockout (*Rictor* fb-KO) mice (see methods). We focused on glutamatergic forebrain neurons because: (a) enhanced excitation of pyramidal neurons is found in chronically

injured epileptogenic neocortex[26], (b) deletion of negative regulators of mTORC1[23,27–30], or constitutive activation of mTOR[31] in excitatory neurons, leads to spontaneous seizures, (c) selective optogenetic-mediated stimulation of the excitatory hippocampal pyramidal cell population via the CamKIIα promoter evokes seizure-like afterdischarges[32], and (d) CamKIIα-mediated expression of inhibitory opsins suppresses train-induced neuronal bursting[33] and seizures in the KA and pilocarpine models of TLE[34,35].

We measured behavioral seizures (Supplementary Fig. 2 and see methods) and electroencephalogram (EEG) patterns induced by KA in WT mice and mice that had mTORC2 genetically silenced only in excitatory forebrain neurons[36]. Strikingly, we found that *Rictor* fb-KO mice, which exhibit a selective reduction in mTORC2 (Fig. 2a–c, Supplementary Fig. 3c, d), but not mTORC1, activity (Supplementary Fig. 3e, f), were more resistant to KA-induced behavioral seizures (Fig. 2d, e). Specifically, *Rictor* fb-KO mice required approximately double the dose of KA (~50 mg/kg) to induce stage 4 behavioral seizures compared to littermate controls (Fig. 2d, e). Moreover, we found that mTORC2-deficient mice injected with KA (25 mg/kg) showed a marked reduction in EEG seizures compared to littermate controls (Fig. 2f, g). Thus, genetic inhibition of mTORC2 protects against KA-induced seizures.

### ASO-mediated inhibition of mTORC2 suppresses acute seizures

In a more therapeutically oriented approach, we used a specific Rictor antisense oligonucleotide (Rictor-ASO) that selectively inhibits mTORC2 activity in mice[23] (Fig. 3a–c, Supplementary Fig. 4). In support of the genetic findings, we found that a single injection with Rictor-ASO reduced acute KA-induced behavioral and EEG seizures (Fig. 3d–g). Thus, ASO-mediated inhibition of mTORC2 was sufficient to reduce acute KA-induced seizures.

Given that human epilepsy is characterized by a growing list of genetically diverse underlying etiologies[7,8,11], we next wondered whether Rictor-ASO could also suppress seizures that arise from distinct mechanisms. To address this question, we employed the pentylenetetrazol (PTZ; a GABAergic antagonist) seizure model[37] (see methods), which we found activated both mTOR complexes (Supplementary Fig. 5a–c). Interestingly, Rictor-ASO-mediated inhibition of mTORC2 reduced PTZ-induced seizures compared to control-ASO (Supplementary Fig. 5d, f). Taken together, our results show that inhibition of mTORC2 reduces acute seizures induced by either KA or PTZ.

### ASO-mediated inhibition of mTORC2 suppresses spontaneous recurrent seizures

To further test the generality of the hypothesis that ASO-mediated inhibition of mTORC2 can suppress seizures across a wide range of causative etiologies, we studied two genetic models of spontaneous recurrent seizures: *Kcna1*-null mice and a "humanized" *MTOR* gain-of-function model of epilepsy. Germline loss-of-function mutations in the *KCNA1* gene, which encodes the voltage-gated potassium channel subunit Kv1.1, are associated with epilepsy in humans[38]. Moreover, *Kcna1*-null mice develop spontaneous seizures that last throughout their life and are a well-established channelopathy model of genetic epilepsy[39]. Interestingly, despite the fact that genetic deletion of *Kcna1* did not modulate the activity of the mTOR complexes compared to WT littermates (Supplementary Fig. 6a–c), treatment with Rictor-ASO significantly reversed their spontaneous recurrent seizures (Fig. 4a–c and Supplemental Fig. 6d, e).

Somatic mutations that activate mTOR have been associated with focal cortical dysplasia, the most common form of focal pharmacoresistant epilepsy in children[11]. Indeed, *in-utero* electroporation (IUE) of activating human *MTOR* variants in mice is sufficient to cause spontaneous seizures, which are attenuated by chronic rapamycin treatment[40]. We examined mice generated by intraventricular IUE of the S2215F gain-of-function MTOR mutation, one of the most frequent

*MTOR* variants associated with human epilepsy[41], and verified that it leads to spontaneous recurrent seizures (Fig. 4d–f). mTORC1 activity was selectively increased on the ipsilateral side of the electroporation compared to the contralateral side (Supplementary Fig. 6f–h). We found that ASO-mediated inhibition of mTORC2 after seizure-onset also drastically reduced seizures in this humanized mTORopathy model (Fig. 4d–f and Supplementary Fig. 6i, j). Taken together, these findings show that inhibition of mTORC2 with Rictor-ASO reverses spontaneous recurrent seizures in two clinically relevant monogenetic mouse models of human epilepsy.

### Unbiased phosphoproteomics analysis identifies a target of mTORC2 implicated in seizures

While TOR was discovered more than 30 years ago[42] and our understanding of mTOR signaling keeps growing[10], little is known about how dysregulated mTOR signaling leads to epilepsy. To gain insights into the potential mechanism by which inhibition of mTORC2 reduces seizures, we utilized an unbiased phosphoproteomics analysis. To this end, we used the *Pten* model of epilepsy because (a) *Pten* fb-KO mice develop spontaneous recurrent seizures and (b) genetic inhibition of mTORC2 (*Pten-Rictor* fb-DKO), but not mTORC1 (*Pten-Rptor* fb-DKO),

reduced seizures arising from *Pten* deficiency[23]. To identify potential mTORC2 substrates driving seizures, we focused on proteins with reduced phosphorylation in *Pten-Rictor* fb-DKO compared to *Pten-Rptor* fb-DKO mice and those that contain a mTORC2 motif, dubbed TOR Interaction Motif (TIM), which has been identified in canonical mTORC2 substrates[43].

We performed global quantitative phosphopeptide profiling on 4-6 independent biological replicates of cortex tissue isolated from 6-week-old mice of the different genotypes. Briefly, we mapped spectra to mouse reference protein sequences and derived a set of high-confidence sequence matches [false discovery rate (FDR) ~1% at both protein and peptide levels] corresponding to 50381 phosphopeptides and 5762 non-phosphorylated peptides (see methods for full description). It is noteworthy that our phosphoproteomics approach identified proteins implicated in epilepsy and neurological dysfunction (Fig. 5a). Moreover, differential phosphorylation analysis identified the canonical mTORC2 targets, including Akt3 (the brain enriched Akt isoform) and conventional PKCs among the most strongly mTORC2-regulated hits (Fig. 5b).

Unexpectedly, we identified Nav1.2 (encoded by Scn2a; Fig. 5b), a key ion channel implicated in epilepsy and neuronal excitability[44–48]

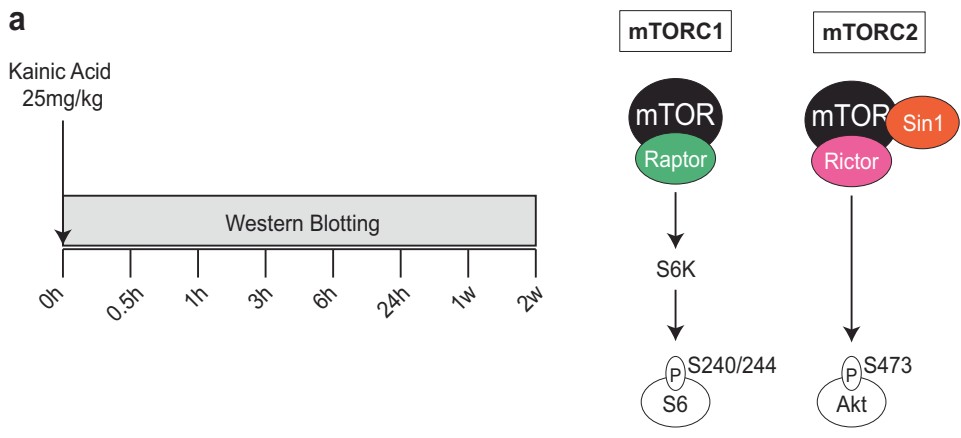

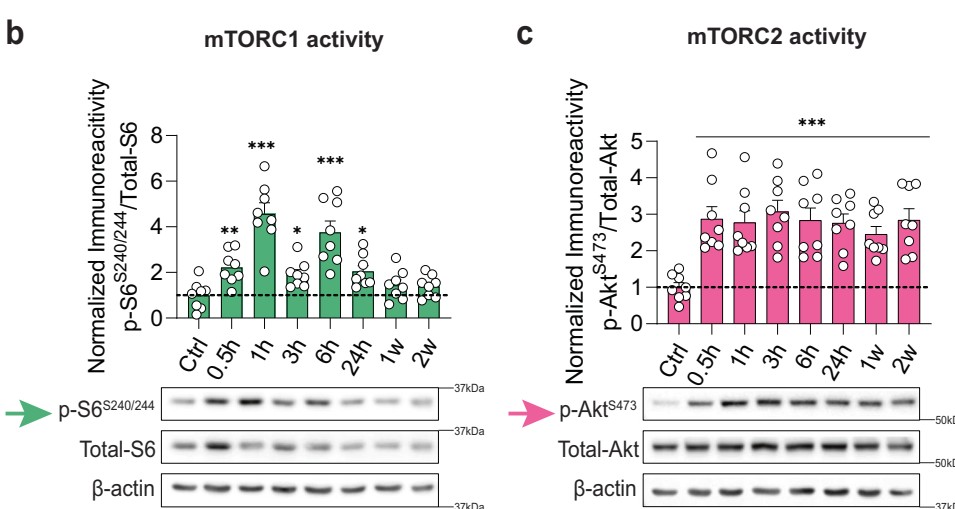

**Fig. 1 | Kainic acid activates mTORC2 more persistently than mTORC1.**
**a** Experimental design and simplified schematic of mTORC1 and mTORC2 signaling. **b, c**, mTOR activity after KA injection. Quantification and representative western blots of mTORC1 activity (in green) measured by p-S6$^{S240/244}$ (**b**, F = 16.05, $P < 0.001$) and mTORC2 activity (in magenta) measured by p-Akt$^{S473}$ (**c**, F = 5.335, $P = 0.0001$)

in the cortex of control (WT) mice ($n = 8$ per group, each timepoint is compared to Ctrl). Data were normalized to naive (ctrl) mice. Statistical tests used for analysis were ordinary one-way ANOVA with uncorrected Fisher's LSD test for multiple comparisons at a 95% confidence interval. Data are represented as means ± s.e.m. *$P < 0.05$, **$P < 0.01$, ***$P < 0.001$. Source data are provided as a Source Data file.

## Genetic inhibition of mTORC2

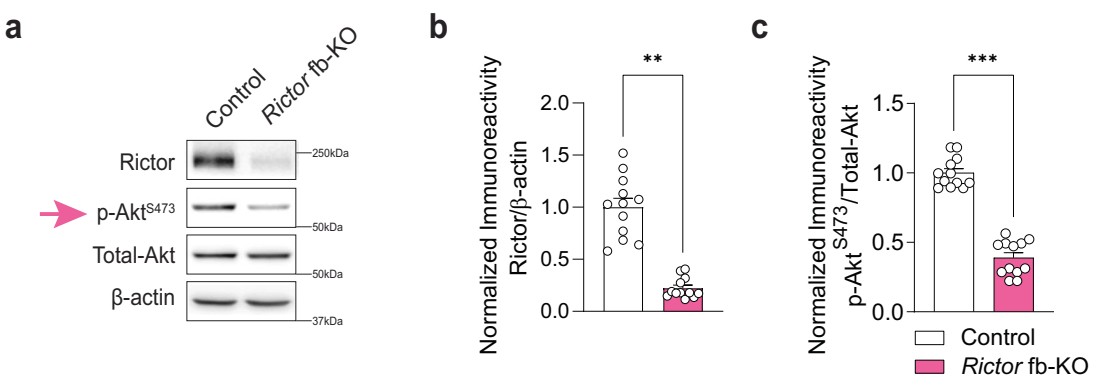

## Behavioral Seizures

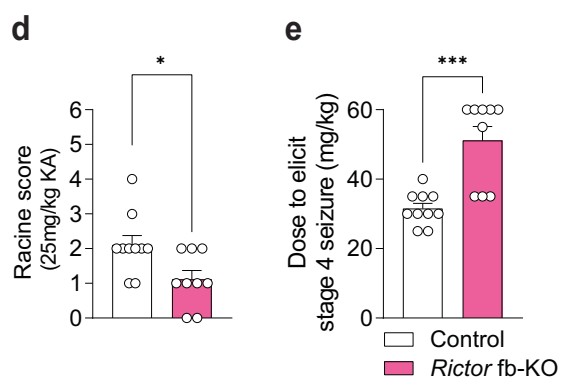

## EEG Seizures (25mg/kg KA)

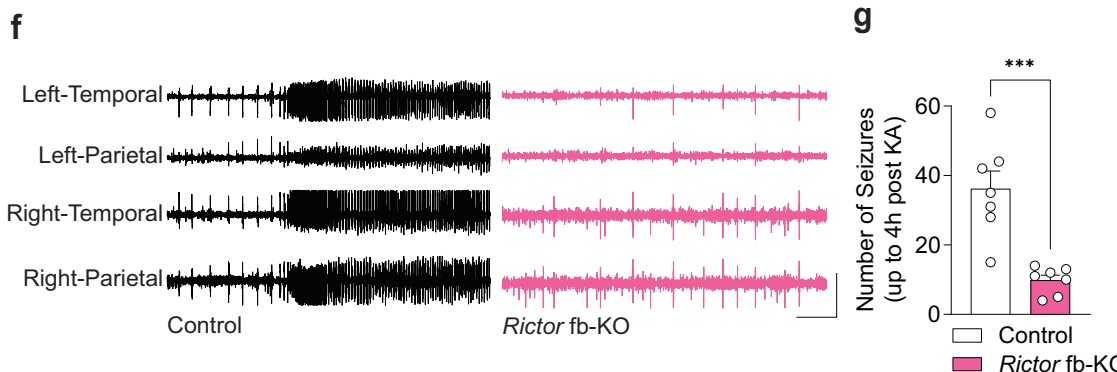

**Fig. 2 | Genetic inhibition of mTORC2 reduces KA-induced seizures.** mTORC2 activity in *Rictor* fb-KO mice. Representative western blots (**a**) and quantification of Rictor (**b**, $t = 4.451$, $P = 0.004$) and p-Akt$^{S473}$ (**c**, $t = 9.210$, $P < 0.001$) in the cortex of control and *Rictor* fb-KO mice ($n = 12$ per group). Behavioral seizures in control and *Rictor* fb-KO mice upon KA injection. Racine score (**d**, U = 19, $P = 0.026$) and dose to elicit stage 4 seizure (**e**, U = 7.5, $P < 0.001$) in control ($n = 10$) and *Rictor* fb-KO ($n = 9$) mice. EEG seizures in control and *Rictor* fb-KO mice after 25 mg/kg KA injection.

Representative EEG traces (**f**) and number of EEG seizures (**g**) from control and *Rictor* fb-KO mice ($n = 7$ per group, $t = 4.906$, $P < 0.001$). Scale bars: X = 10 s, Y = 1 mV. Statistical tests used for analysis were two-tailed unpaired *t*-tests for normally distributed data or Mann-Whitney test for non-normally distributed data at a 95% confidence level. Data are represented as means ± s.e.m. *$P < 0.05$, **$P < 0.01$, ***$P < 0.001$. Source data are provided as a Source Data file.

among the highest-ranked mTORC2-regulated phosphorylated proteins. For mTORC2 to regulate Nav1.2 phosphorylation, mTORC2 would have to localize to the plasma membrane, as it has been previously demonstrated[49], and Nav1.2 should selectively interact with the mTORC2 complex.

mTORC1 and mTORC2 share two major components, mTOR and the mammalian lethal with SEC13 protein 8 (mLST8). The

defining core subunit of mTORC1 is Raptor, whereas the defining subunits of mTORC2 are the adaptor protein Rictor and the scaffolding protein Sin1, which is believed to harbor the substrate binding site[50–52]. Like the canonical targets of mTORC2, we predicted that Nav1.2 would specifically interact with mTORC2 (but not mTORC1). To test this prediction, we expressed GFP-Scn2a in HEK293T cells and assessed its binding to the endogenous defining

## ASO-mediated inhibition of mTORC2

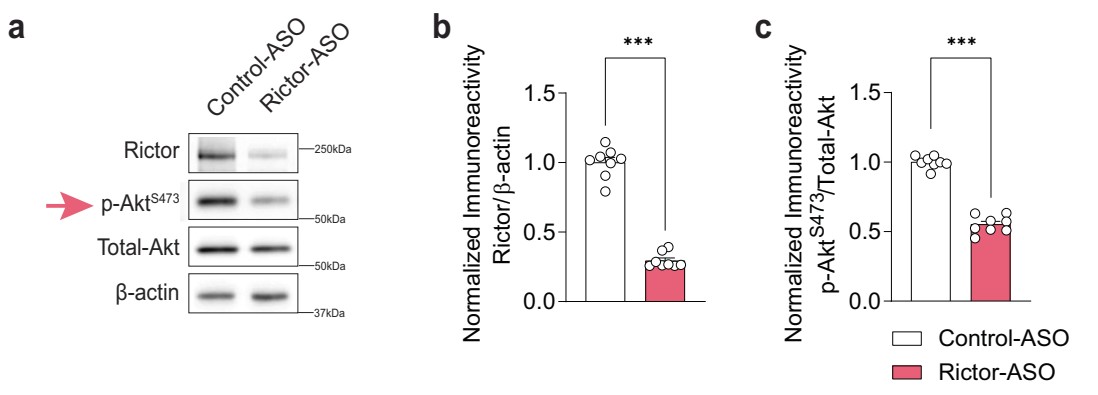

## Behavioral Seizures

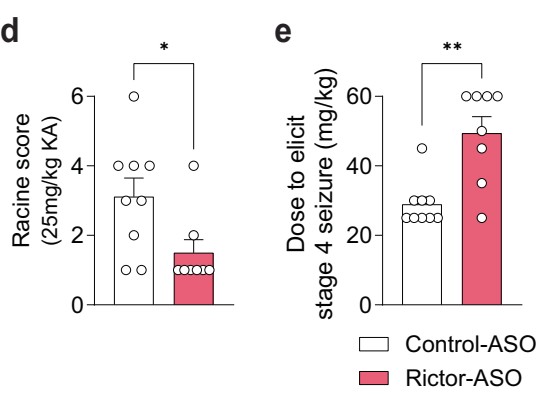

## EEG Seizures (25mg/kg KA)

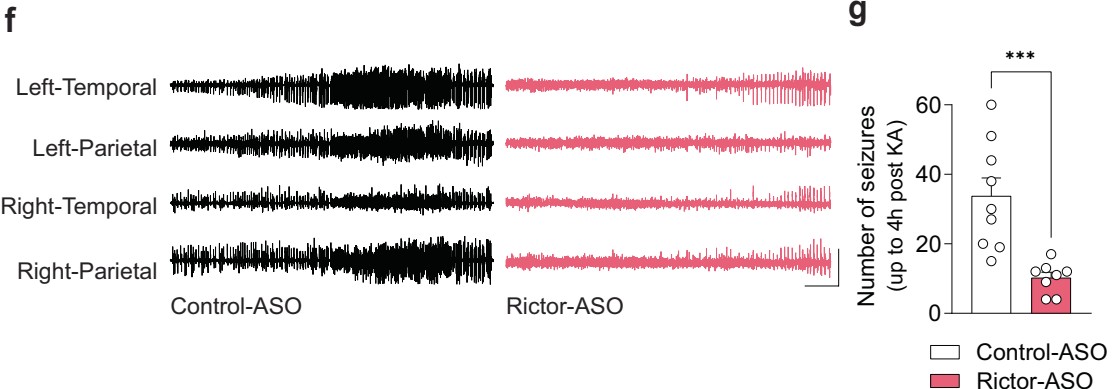

**Fig. 3 | ASO-mediated inhibition of mTORC2 reduces KA-induced seizures.** mTORC2 activity in Rictor-ASO treated naïve mice. Representative western blots (**a**) and quantification of Rictor (**b**, $t = 5.016$ $P < 0.001$) and p-Akt$^{S473}$ (**c**, U = 1, $P < 0.001$) in control-ASO and Rictor-ASO treated mice ($n = 8$ per group). KA-induced behavioral seizures in Rictor-ASO treated mice. Racine score (**d**, U = 15, $P = 0.033$) and dose to elicit stage 4 seizure (**e**, U = 8, $P = 0.004$) in control-ASO ($n = 9$) and Rictor-ASO ($n = 8$) treated mice. KA-induced acute EEG seizures in control-ASO ($n = 9$) and Rictor-ASO ($n = 8$) treated mice ($t = 4.059$, $P < 0.001$). Representative EEG traces (**f**) and number of EEG seizures (**g**) from control-ASO and Rictor-ASO treated mice. Scale bars: X = 10 s, Y = 1 mV. Statistical tests used for analysis were two-tailed unpaired t-tests for normally distributed data or Mann-Whitney test for non-normally distributed data at a 95% confidence level. Data are represented as means ± s.e.m. *$P < 0.05$, **$P < 0.01$, ***$P < 0.001$. Source data are provided as a Source Data file.

components of mTORC2 or mTORC1 (Rictor and Raptor, respectively). Remarkably, and consistent with our prediction, we found that GFP-Nav1.2 pulled-down endogenous rictor, but not raptor (Fig. 5c). In a complementary approach, immunoprecipitation of HA from cells co-expressing GFP-Scn2a and HA-Sin1 showed that Sin1 binds to Nav1.2 (Fig. 5d). Collectively, these findings identify Nav1.2,

which is a key regulator of neuronal excitability and seizures[46,48], as a selective target of mTORC2.

## Discussion

Elucidating common molecular pathways driving epilepsy amidst the growing complexity of monogenic epilepsies has been challenging. In

## *Kcna1*-null channelopathy model

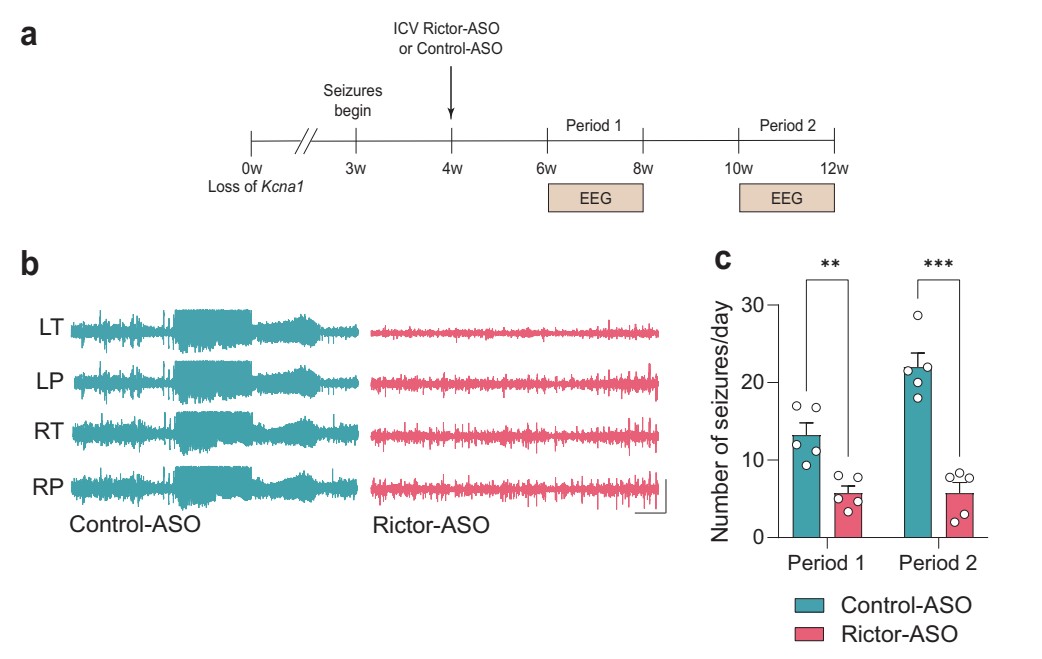

## *MTOR*^S2215F gain-of-function model

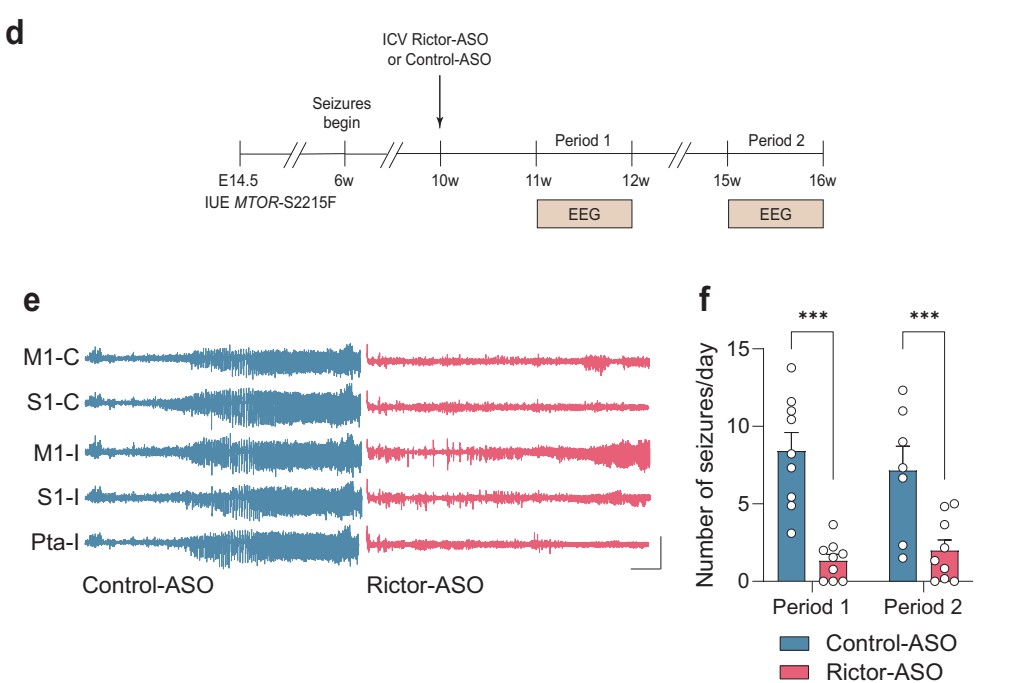

**Fig. 4 | ASO-mediated inhibition of mTORC2 reduces seizures in two preclinical models of epilepsy.** EEG seizures in *Kcna1*-null mice treated with either control-ASO or Rictor-ASO (n = 5 per group). Treatment schematic (**a**), representative EEG traces (**b**) and number of EEG seizures per day (**c**) from control-ASO and Rictor-ASO treated *Kcna1*-null mice. Period 1: 6–8-week-old mice (P = 0.034), Period 2: 10–12-week-old mice (P < 0.001). P: Parietal Cortex, T: Temporal Cortex, L: Left, R: Right. EEG seizures in *MTOR*^S2215F mice treated with either control-ASO or Rictor-ASO. Treatment schematic (**d**), representative EEG traces (**e**) and number of EEG seizures

per day (**f**) in control-ASO (n = 9, 7) and Rictor-ASO (n = 9, 9) treated *MTOR*^S2215F mice. Period 1: 11–12-week-old mice (P < 0.001), Period 2: 14–15-week-old mice (P < 0.001). M1: Primary Motor Cortex, S1: Primary Somatosensory Cortex, Pta: Parietal Cortex, C: Contralateral, I: Ipsilateral. Scale bars: X = 10 s, Y = 1 mV. Statistical tests used for analysis were two-tailed unpaired t-tests for normally distributed data or Mann-Whitney test for non-normally distributed data at a 95% confidence level. Data are represented as means ± s.e.m. *P < 0.05, **P < 0.01, ***P < 0.001. Source data are provided as a Source Data file.

addition, attempts to prevent epileptogenesis, the gradual appearance of epilepsy during brain development, by modulating key molecular pathways have yielded limited success. Our convergent findings, which

combine molecular genetics with ASO-based technology, demonstrate that inhibition of mTORC2 both prevents and reverses the epileptic phenotype in a wide range of seizure disorders of disparate underlying

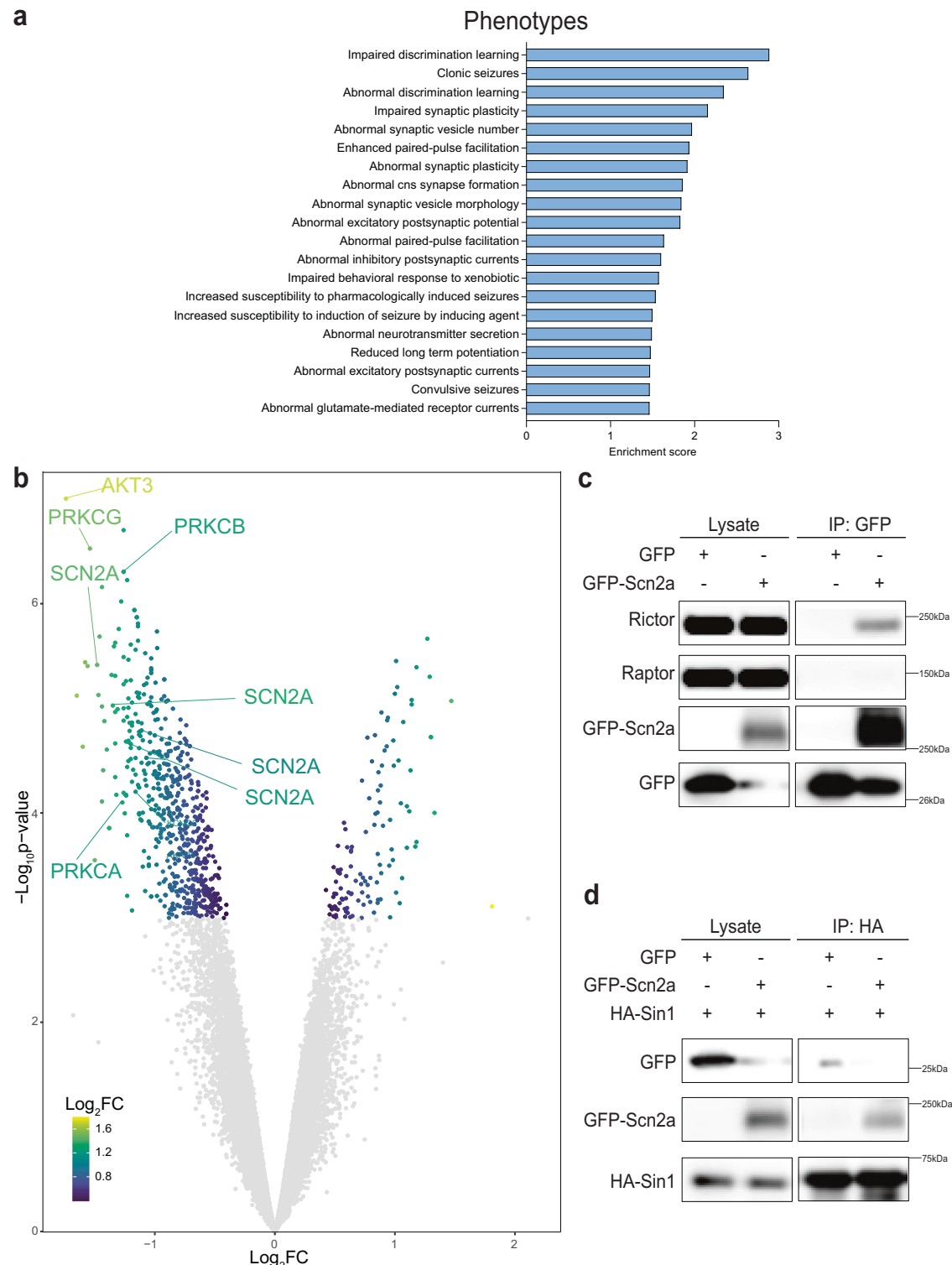

**Fig. 5 | Nav1.2 is regulated by Rictor and binds to the mTORC2 components Rictor and Sin1. a** Volcano plot showing differentially regulated phosphorylation events in *Pten-Rictor* fb-DKO (*n* = 6) compared to *Pten-Rptor* fb-DKO (*n* = 4) samples (Threshold *P* < 0.05). **b** Pathway enrichment analysis for phenotype (Monarch MPO) and cellular component (GO component) in downregulated phosphorylation events in *Pten-Rictor* fb-DKO compared to *Pten-Rptor* fb-DKO samples (FDR < 1%).

Pull-down assays from HEK 293 T cells expressing GFP or GFP-Scn2a alone and co-expressed with HA-Sin1. **c** Representative western blots of lysates and pull-downs for GFP, Raptor, Rictor and β-actin from GFP pull-downs. **d** Representative western blots of lysates and pull-downs for GFP, HA, Raptor, Rictor and β-actin from HA pull-downs. Data shown are representative of 3 independent experiments. Source data are provided as a Source Data file.

etiologies (Figs. 2d–f, 3d–f, 4 and Supplementary Figs. 5, 6). These findings are unexpected since hyperactivation of mTORC1 has been proposed as the leading cause of epilepsy in mTORopathies and other seizure disorders[12,19]. Moreover, Rictor-ASO reduces seizures in multiple models of epilepsy regardless of if mTORC2 is altered. Thus, our findings support the notion that inhibition of mTORC2, which is structurally and functionally different from mTORC1, suppresses seizures arising from various etiologies by buffering the pathological hypersynchrony during epilepsy.

The contributions of the mTOR complexes to seizures are not fully understood. It is noteworthy that our data do not rule out a role for mTORC1 in seizure disorders. Indeed, we believe that in function of the upstream mTORopathy causing mutation, mTORC1 or mTORC2 may be relevant. For instance, in *TSC, DEPDC5* or *STRADA* related seizure disorders, mTORC1 is hyperactive and mTORC2 activity is often reduced or unchanged, indicating that mTORC1 may be the major driver. Incidentally, a recent report shows that partial deletion of *Rptor* rescued *Tsc1*-related phenotypes[53]. Deletion of *Rptor* in *Tsc1*-deficient mice not only reduced hyperactive mTORC1, but also enhanced the reduced mTORC2 activity. This increase in mTORC2 activity upon *Rptor* deletion is due to the absence of the inhibitory feedback onto IRS1-PI3K, which is an inherent feature of mTOR signaling[54–57]. Given that deletion of *Rptor* rebalances both mTORC1 or mTORC2 activity in the *Tsc1*-deficient model, it remains unclear whether the rescued phenotype is due to normalization of either mTORC1 or mTORC2 activity. In contrast, here we show that inhibition of mTORC2 suppresses seizures in several models of epilepsy. Thus, we propose that seizure disorders should be stratified based on their functional mTORC1 or mTORC2 dependency.

Whether inhibition of mTORC2 could suppress all types of seizure disorders, including those emerging because of a head injury, infectious agents, tumors, or vascular anomalies remains to be determined. We previously found that genetic inhibition of mTORC2 in forebrain (CamKIIα-Cre positive) neurons enhanced lifespan and suppressed seizures in a very robust and fully penetrant *Pten*-deficient model of epilepsy (with 100% mortality by 12 weeks of age)[23]. In contrast, when *Pten* was deleted from GFAP expressing cells, mTORC2 inhibition failed to reduce seizures[58]. It is noteworthy that this model is highly dependent on the genetic background (only ~22% of the *Pten*-deficient mice exhibited rare seizures with >50% of mice surviving up to 20 weeks of age) and the study did not determine whether Rictor levels are reduced in the *Pten-Rictor* deficient mice[58]. Taken together, it is possible that silencing mTORC2 in different cell types (CamKIIα-Cre positive vs GFAP-Cre positive) could have different effects on seizures.

While mTOR dysregulation has been associated with epilepsy[11], the mechanism by which hyperactivation of mTOR signaling leads to seizures remains elusive. Here, we show that mTORC2 modulates the phosphorylation of Nav1.2, a key protein linked to seizures (Fig. 5b)[44–48]. Moreover, Nav1.2 binds to mTORC2, but not mTORC1 (Fig. 5c, d), demonstrating the specificity of this interaction. Nav1.2 is highly regulated by phosphorylation[59,60]. While the mTORC2-regulated phosphorylation sites identified here on Nav1.2 are novel, they span two important domains within the protein: the I-II linker domain that regulates sodium current and favors slow inactivation[59] and the C-terminal region that regulates degradation of Nav1.2[61]. Given that both gain-of-function and loss-of-function mutations in Nav1.2 have been implicated in epilepsy[46,48], in future experiments, it would be interesting to examine the role of these mTORC2-mediated phosphorylation events on channel function.

While mTORC2-mediated phosphorylation of Nav1.2 may be a potential mechanism of seizure inductions, our phosphoproteomics analyses identified additional proteins related to synaptic transmission and epilepsy in the brain that are likely modulated by mTORC2 (Fig. 5a). Thus, persistent hyperactivation of mTORC2 could lead to seizures by phosphorylating several targets, which in concert could promote abnormal network rhythmicity. This hypothesis is consistent with mTOR complexes driving the phosphorylation of multiple targets to regulate specific cellular processes. For instance, mTORC1 regulates protein homeostasis (proteostasis) by phosphorylation of S6Ks and 4EBPs and regulating protein synthesis, while simultaneously regulating autophagy by phosphorylation of ULK1 and TFEB[10]. The functional validation and the contribution of different mTORC2 candidates to the seizure phenotype (or their combinatory effect), while difficult to test experimentally, could uncover novel seizure pathways.

Finally, ASOs are increasingly recognized as promising therapeutic agents as illustrated by effective treatments of Duchenne muscular dystrophy and spinal muscular atrophy using ASOs[62]. Indeed, ASO-based clinical trials to treat central nervous systems disorders, including amyotrophic lateral sclerosis (ALS), Parkinson's, Huntington's and Alzheimer's disease, are currently underway[63]. Of future clinical relevance and as a promising alternative to small molecules, is our ASO-mediated therapy (Rictor-ASO), which selectively inhibits mTORC2 and suppressed seizures in several models of epilepsy. This strategy is particularly pertinent to targeting mTORC2, since potent and selective small molecule mTORC2 inhibitors that spare mTORC1 activity remain elusive[64,65]. The deletion of *Rictor* during development has a significant effect on brain and body size[66,67]. In contrast, when *Rictor* was deleted from forebrain neurons postnatally (*Rictor* fb-KO mice), there were no gross brain abnormalities, changes in the expression of several synaptic markers or alterations in basic synaptic properties in *Rictor* fb-KO mice[68]. Accordingly, we found that Rictor-ASO is safe and tolerable[23] and did not affect activity levels or learning and memory (Supplementary Fig. 7). In light of this, our findings support the interesting possibility that treatment with Rictor-ASO could broadly suppress seizures with different underlying etiologies in humans.

## Methods

### Mouse husbandry

Experiments were conducted on 8–12-week-old male and female mice with the C57Bl/6 background unless otherwise stated. *Rictor*^loxP/loxP^ mice have been described previously[23,68,69]. *Kcna1*-null mice have been previously described and were kept on the C57Bl/6 background[39]. Mice were weaned at the third postnatal week and genotyped by polymerase chain reaction (PCR). *Cre* expression, *Rictor* and wild-type alleles were detected by PCR as previously described[68,69]. *Kcna1* mutant and wild-type alleles were detected by PCR assay with two pairs of primer sets as previously described[39]. Mice were kept on a 12 h/12 h light/dark cycle (lights on at 7:00) and had access to food and water *ad libitum*. Animal care and experimental procedures were approved by the institutional animal care and use committee (IACUC: AN5068) at Baylor College of Medicine, according to NIH guidelines.

### Behavioral seizure monitoring and scoring

Mice were injected intraperitoneally (i.p.) with 25 mg/kg kainic acid (KA; Sigma), and behavioral seizures were monitored every 30 minutes for a period of 4 hours using the following modified Racine scale: 0, no response; 1, behavioral arrest/motionless staring; 2, head nodding; 3, forelimb clonus; 4, forelimb clonus with rearing and falling; 5, generalized tonic-clonic activity with wild jumping often resulting in death; 6, death. If mice exhibited a stage 4 behavioral seizure during the first 30 minutes, no further KA was administered. Otherwise, mice were progressively injected with a low dose of KA (5 mg/kg) every 30 minutes for 4 hours until they reached stage 4.

### EEG recordings

Mice were anesthetized with isoflurane and Teflon-coated silver wire electrodes were implanted bilaterally into the subdural space over the frontal, temporal, and parietal cortices. After a 72-hour post-surgical recovery period, freely moving mice were recorded for 4 hours post-

injection in the KA-model or continuously for 24 hours in the spontaneous seizure models (*Kcna1*-null and *MTOR*[S2215F]). Digitized video electroencephalographic (EEG) data were obtained daily for two weeks from mice. All EEG data including seizure frequency was interpreted by analysts with expertise in epilepsy who were blinded to genotype. Recordings were visually inspected in their entirety. EEG signals were sampled at 2 kHz and a band pass filter (0.5–500 Hz) was applied during recording. A seizure was defined by high amplitude activity >2-fold the background, with a frequency >5 Hz and a minimum duration of 10 seconds. Electrographic seizure activity was verified by inspection of the concurrent video with documentation of motor seizure episodic behavior consisting of arrest, clonic limb movements, tonic posture and tonic elevation of the tail (Straub tail).

### PTZ paradigm
The PTZ kindling paradigm was performed as previously described[37]. Briefly, mice were injected i.p. with 50 mg/kg PTZ every other day over 10 days. After each injection, EEG data was recorded for 1-hour post-PTZ and the number of seizures was quantified.

### Behavioral tests
Similar numbers of male and female 8-week-old mice were used for behavior experiments. To control for odor cues, apparatus was cleaned with ethanol, dried and ventilated between testing of individual mice. Mice were handled for 5 consecutive days (15 minutes/day) before behavioral experiments. Littermates were used for all behavioral experiments and mice were tested at the same time daily during the light cycle with the experimenter blinded to treatment group.

**Open field test.** Mice were placed in an open arena (40 × 40 cm, height 20 cm) and allowed to explore freely for 10 minutes and their position was monitored using AnyMaze tracking software. Distance travelled, speed and time spent in the center (inner zone) and periphery (outer zone) of the arena were recorded and measured throughout the task. The center of the arena was defined as the inner 20 × 20 cm area.

**Novel object recognition.** Mice were habituated to a black Plexiglass rectangular chamber (31 × 24 cm, height 27 cm) for 20 minutes under dim ambient light for three days. On the fourth day, test mice were allowed to explore two identical objects for 10 minutes then returned to the home cage. 24 hours later, the mice were returned to the chamber and presented with a familiar object used on the previous day, and a novel object of equal height and volume, but different shape and appearance for 10 minutes. Exploration was defined as sniffing (with nose contact or head directed to the objects) within a 2 cm radius of the object. Object recognition was determined by the discrimination index which was computed as [(novel object exploration time-familiar object exploration time)/total exploration time] × 100.

### Western blotting
Western blotting was performed as we previously described[23]. Mice were euthanized in isoflurane, hippocampus and cortex tissue was isolated, homogenized in cold lysis buffer 200 mM HEPES, 50 mM NaCl, 10% glycerol, 1% Triton X-100, 1 mM EDTA, 50 mM NaF, 2 mM Na₃VO₄, 25 mM β-glycerophosphate and EDTA-free complete ULTRA tablets (Roche, Indianapolis, IN), and centrifuged at 13,000 $g$ for 20 minutes. Supernatants were collected and protein concentrations were measured using a Bradford Assay. 20 μg of protein was loaded per sample, resolved on SDS-PAGE (6–10%) and transferred onto nitrocellulose membranes (BioRad). Membranes were blocked in 5% milk in TBST for 1 hour before overnight incubation in primary antibodies. Membranes were then washed 4 times in TBST and blocked in secondary antibodies (Proteintech #SA00001-2 or #SA00001-1, 1:5000) diluted in 5% milk in TBST for 1 hour. Blocked membranes were washed 5 times in TBST and then developed using an imager (LICOR

Odyssey or Azure 600). Phospho-antibodies and total antibodies were blotted on the same membranes after incubation in stripping buffer (200 mM Glycine, 0.1% SDS, 1% Tween 20 v/v, pH 2.2) for 1 hour. Antibodies against Raptor (1:1000, #2280), Rictor (1:1000, #2114), p-S6 (1:2000, Ser240/244, #5364), p-Akt (Ser473, 1:1000, #9271), p-PKC (S657, 1:2000 #9371), p-NDRG1 (T346, 1:1000 #3217), PKCα (1:1000 #2056), Total-NDRG1 (1:1000 #9408) Total-S6 (1:1000, #2217), Total-Akt (1:1000 #9272), GFP (1:1, 000 #2956), HA (1:1000 #3734) and β-actin (1:10,000 #3700) were purchased from Cell Signaling Technology (Danvers, MA).

### DNA constructs and plasmids
HA-Sin1 was a generous gift from Dr. Alexandra Newton (University of California, San Diego). GFP-Scn2a and GFP were synthesized and sequenced by Epoch Life Science, Inc. (Missouri City, Texas).

### Cell culture and transfection
HEK293T cells (ATCC #CRL-1573) were grown in DMEM (Sigma #D5796) supplemented with 10% FBS (Gibco #10270106) at 37 °C and 5% CO₂. Transfection was performed with Lipofectamine 3000 (ThermoFisher Scientific # L3000008) following manufacturer's recommendation.

### Immunoprecipitation
Cells were homogenized in ice-cold CHAPS lysis buffer (40 mM HEPES (pH 7.5), 120 mM NaCl, 1 mM EDTA, 10 mM glycerophosphate, 50 mM NaF, 1.5 mM Na3VO4, 0.3% CHAPS (wt/vol) and EDTA-free complete ULTRA Tablets (Roche)). Lysates were rotated at 4 °C for 20 mins and then centrifuged at 13,000 $g$ for 20 min. 25 μL of the indicated antibodies attached to magnetic agarose beads (ChromoTek GFP-Trap Magnetic Particles M-270, proteintech #gtd; Pierce Anti-HA Magnetic Beads, ThermoFisher Scientific #88836) was equilibrated in lysis buffer, added to the supernatant and incubated with rotation at 4 °C for 1 h or overnight. Immunoprecipitates were washed four times with lysis buffer and samples were resolved by SDS-PAGE (7.5–10%) and immunoblotted with specific primary antibodies.

### Antisense oligonucleotide synthesis
ASOs consist of 20 chemically modified nucleotides, five 2′-O-methoxyethyl-modified nucleotides at each end separated by ten DNA nucleotides in the center. The backbone of the ASOs consists of a mixture of modifications from 5- to 3-: 1-PS (phosphorothioate), 4-PO (phosphodiester), 10-PS, 2-PO and 2-PS. Rictor-ASO (GTTCACCCTA-TACATTACCA) targeting mouse *Rictor* mRNA and a Control-ASO (CCTATAGGACTATCCAGGAA) were developed and synthesized by Ionis Pharmaceutical as previously described[23]. The algorithm Bowtie44 was used to determine potential off-targets for the Rictor-ASO and confirmed that Rictor-ASO binds with 100% complementarity (zero mismatch) to the mouse Rictor transcript and does not bind to any other mRNA with full complementarity in the mouse transcriptome.

### Intracerebroventricular ASO injection
The surgical site was sterilized with betadine and 70% alcohol. Buprenorphine (1 mg/kg) was administered subcutaneously 1 hour before surgery for pain control. Mice were anesthetized with 3% isoflurane for 10 minutes before placing on a computer-guided stereotaxic instrument (Kopf Instruments) fully integrated with the Franklin and Paxinos mouse brain atlas through a control panel. Anesthesia was continuously delivered (2% isoflurane) throughout the surgery. A midline incision was made on the scalp and a small hole was drilled through the skull above the right lateral ventricle. Rictor-ASO (500 μg) was diluted in PBS and then ICV delivered (10 μL) using a Hamilton syringe and glass needles. Control mice received control-ASO (500 μg). The coordinates used for ICV injection were as follows: anteroposterior (AP) = − 0.2 mm, mediolateral (ML) = 1 mm, dorsoventral (DV) = −3 mm. The

needle was left for 2 minutes on the site of injection. The incision was manually closed with sutures. Mice were maintained on a 39 °C iso-thermal pad while anesthetized and during recovery.

### In utero electroporation

$MTOR^{S2215F}$ mice were generated by *in utero* electroporation (IUE) of DNA constructs at E14.5 in Swiss/CD1 embryos (Janvier Labs) as previously described[70]. DNA solution contained 0.5 mg/mL of pCAG-EGFP and 2.5 mg/mL pCAGIG-MTOR (p.S2215F, kindly provided by A. Represa team at INMED, Marseille). At birth, pups were selected based on the visualization of a GFP fluorescent stain in the cortex. Mouse studies were approved by the French Ministry of Research (no. APAFIS #37296 and #40207).

### Tissue digestion and phosphopeptide enrichment

Mice were euthanized using isoflurane and cortex tissue was snap frozen in liquid nitrogen, lysed in 500 μL 8 M guanidinium hydro-chloride (GndHCl) and centrifuged at 16300 $g$ in 4°C for 15 minutes after which the supernatants were recovered. 1 mg of protein was digested by adding 100 mM Triethylammonium bicarbonate (TEAB) treated with 8.8 mM Tris (2-carboxyethyl) phosphine hydrochloride at 56°C for 15 minutes, followed by a 30-minute incubation at room temperature in the dark with 15 mM iodoacetamide. Samples were diluted with 100 mM TEAB to reduce the concentration of GdnHCl to 1 M, supplemented with 5% (w/w) trypsin (TPCK, ThermoFisher Scientific), incubated for 12 hours at 37°C, re-supplemented with 5% (w/w) trypsin for another 6 hours. Samples were acidified in formic acid, digests were centrifuged at 16300 $g$ in 4 °C for 15 minutes, super-natants were recovered, and peptides were extracted and desalted using a MAX_RP Sep Pak ® classic C18 cartridge (Waters). TMT labelling was performed on samples using TMTPro™−18 label plex kit (ThermoFisher Scientific, #A52045), following manufacturer's instructions. Phosphopeptide enrichment was performed in an AKTA Purifier (GE Healthcare, Piscataway, NJ) using 5 μm titanium dioxide (TiO₂) beads (GL Sciences, Tokyo, Japan) in-house packed into a 4.0 mm×3.0 cm analytical column (Upchurch Scientific, Oak Harbor, WA). Phospho-peptide enriched samples were fractionated using high pH reverse phase chromatography.

### Mass spectrometry analysis

MS spectra were acquired between 375 and 1500 m/z with a resolution of 120000. For each MS spectrum, multiply charged ions over the selected threshold (2E4) were selected for MSMS in cycles of 3 seconds with an isolation window of 0.7 m/z. Precursor ions were fragmented by HCD using stepped relative collision energies of 30, 35 and 45 in order to ensure efficient generation of sequence ions as well as TMT reporter ions. MSMS spectra were acquired in centroid mode with resolution 60000 from m/z = 120. A dynamic exclusion window was applied which prevented the same m/z from being selected for 30 second after its acquisition.

### Peptide and protein identification and TMT quantitation

Peak lists were generated using PAVA in-house software[71]. All gener-ated peak lists were searched against the mouse subset of the Swis-sProt database (SwissProt.2019.07.31), using Protein Prospector[72] with the following parameters: Enzyme specificity was set as Trypsin, and up to 2 missed cleavages per peptide were allowed. Carbamido-methylation of cysteine residues, and TMTPro16plex labeling of lysine residues and N-terminus of the protein were allowed as fixed mod-ifications. N-acetylation of the N-terminus of the protein, loss of pro-tein N-terminal methionine, pyroglutamate formation from of peptide N-terminal glutamines, oxidation of methionine and phosphorylation on serine, threonine and tyrosine were allowed as variable modifica-tions. Mass tolerance was 6 ppm in MS and 30 ppm in MS/MS. The false positive rate was estimated by searching the data using a concatenated

database which contains the original SwissProt database, as well as a version of each original entry where the sequence has been rando-mized. A 1% FDR was permitted at the protein and peptide level. For quantitation only unique peptides were considered; peptides common to several proteins were not used for quantitative analysis. Relative quantization of peptide abundance was performed via calculation of the intensity of reporter ions corresponding to the different TMT labels, present in MS/MS spectra. Intensities were determined by Protein Prospector. Summed intensity per sample on each TMT channel for all identified spectra were used to normalize individual intensity values. Relative abundances were calculated as ratios vs the average intensity levels in the 3 channels corresponding to control samples. Spectra representing replicate measurements of the same peptide were kept and used to calculate the dispersion and the sig-nificance threshold for the analysis. Non phosphorylated peptides were used to estimate relative protein levels. For total protein relative levels, peptide ratios were aggregated to the protein levels using median values of the log2 ratios.

### Phosphoproteomics data processing and analysis

To control for different protein loading across the TMT channels, cor-rection factors were applied such that the summed protein amount per channel was the same across channels. Data were then log2-transformed. If multiple forms of the exact same phosphopeptide but with differing additional modifications were detected, we kept the most abundant phosphopeptide for the statistical analysis. Statistical ana-lyses were performed using R including data processing and plotting. To identify differentially phosphorylated peptides between any two sets of mice (e.g., *Pten-Rictor* and *Pten-Rptor* DKO), we calculated statistical significance and corrected for multiple hypothesis testing by using a permutation-based false discovery rate approach[73]. Enrichment analysis was performed on the full list of phosphoproteins using Wilcoxon rank test on the fold changes between *Pten-Rictor* and *Pten-Rptor* DKO and applying multiple hypotheses correction as implemented in STRING[74]. To avoid doble-counting of proteins, for each protein the phospho-peptide with the lowest fold change was kept (i.e., the one in which *Pten-Rictor* had the lowest phosphorylation relative to *Pten-Rptor* DKO).

### Statistics and Reproducibility

No statistical methods were used to pre-determine sample sizes, but our sample sizes are selected based on previous studies published in the field[23,27,75–79]. Animals in the same litter were randomly assigned to different treatment groups in various experiments. No animals or data points were excluded from the analysis. Normality testing and F-tests of homogeneity of variances were performed before choosing statis-tical tests. Statistics are based on the two-sided Student's t-test or Mann–Whitney rank sum test for two-group comparisons (for datasets that were not normally distributed). One-way analysis of variance (ANOVA) and the uncorrected Fisher's least significant difference (LSD) method for pairwise comparisons analysis was performed for multiple comparisons, unless otherwise indicated. $P < 0.05$ was con-sidered significant (*$P < 0.05$, **$P < 0.01$, ***$P < 0.001$). Statistical ana-lyses were performed on Prism 9 (GraphPad Software Inc.) or R.

### Reporting summary

Further information on research design is available in the Nature Portfolio Reporting Summary linked to this article.

## Data availability

Source data are provided with this paper. The mass spectrometry proteomics data have been deposited to the ProteomeXchange Con-sortium via the PRIDE partner repository with the dataset identifier PXD045878. The project can be accessed at ftp://ftp.pride.ebi.ac.uk/pride/data/archive/2023/10/PXD045878. Source data are provided with this paper.

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

## Acknowledgements

This research was supported by funding from NIH (R01 NS124145) and the generous support from the Sammons Enterprise to M.C.-M, support from the Dr. Miriam and Sheldon G Adelson Medical Research Foundation to A.B. and funding from the European Research Council (N°682345 and ANR-10- IAIHU-06) to S.B. We thank all the members of the Costa-Mattioli lab and Delphine Roussel from the ICM PHENO-ICMice facility for their input in preparing this manuscript.

## Author contributions

J.O. and M.C.-M. conceived and planned the experiments. J.O., J.M., Alex.B., J.O.P., S.T., C.-J.C., K.I., M.D. and H.Z. performed the experiments and analyzed the data. J.O., J.M., Alex.B., H.Z., M.D., K.I., J.L.N. and S.B. contributed to the seizures and ASO study design and analysis. J.O.P., S.T. and Alma.B, contributed to the mass spectrometry and phosphoproteomics study design and analysis experiments. J.O. and M.C.-M wrote the manuscript, with contributions from J.M., Alex.B., P.J.-N., S.B. J.O.P. and J.L.N.

## Competing interests

M.C.-M., J.O., S.T. and H.Z. are employees of Altos Labs, Inc. M.C.-M is a shareholder and an employee of Altos Labs, Inc. and a shareholder of Mikrovia, Inc. All other authors declare no competing interests.
