## [Peer Review File · Nature Communications]

Targeted suppression of mTORC2 reduces seizures across models of epilepsyEditorial Note: Parts of this Peer Review File have been redacted as indicated to remove third-party material where no permission to publish could be obtained.

REVIEWER COMMENTS

Reviewer #1 (Remarks to the Author):

In this submission, Okoh et al. study whether reducing expression of the Rictor gene, either with Cre-lox technology or an ASO, can lessen either acute or spontaneous seizure burden in four mouse models of epilepsy. Rictor is a key component of the mTORC2 complex, and the authors show evidence that Rictor reduction reduces phosphorylation of Akt at the mTORC2 site. Rictor deletion via CamKII promoter or the Rictor ASO reduced the number of acute seizures after KA administration. The Rictor ASO also reduced the number of spontaneous seizures in the Kcna1 mouse model and a mouse with IUE overexpressing an mTOR mutant.

The strength of this paper is that it is interesting that Rictor downregulation affects seizure frequency in a number of different epilepsy models, both acute and chronic. The authors previously showed that Rictor downregulation suppressed seizures in a Pten model of epilepsy, so this is a nice follow up to that paper showing that the effect is not specific to the Pten model. The main conclusion of the paper, however, that mTORC2 “drives” epilepsy in these models, is not supported. Also, there is a general lack of experimental detail and depth. These contribute to a lack of insight into why mTORC2 inhibition may protect the brain from both acute and chronic epilepsy, and whether it would be a viable drug target.

Specific points.

There is no evidence that mTORC2 is driving or playing a role in the development of seizures in these models. Other approaches to inhibit brain activity, like treating with barbiturates, would also likely suppress these seizures, but I wouldn't interpret that data as suggesting that GABA receptor dysfunction is causing seizures in these models. The paper should not make claims that mTORC2 activation causes seizures or plays a role in their development.

The finding that Rictor loss lessens the severity of the acute KA seizures suggests that mTORC2 inhibition is more a general suppressor of brain excitability than a participant in epileptogenesis.

It is well known that mTORC2 activates mTORC1 via Akt-Tsc signaling. It is therefore possible that the protective effect of Rictor downregulation occurs via inhibition of mTORC1. This is not explored experimentally or discussed.

There is a lack of depth in the reporting of the results that seems like a missed opportunity to achieve further insight. With the use of Western blots, it shouldn't be too difficult to examine other targets of mTORC1 and mTORC2, the timeline or presence of mTORC2 activation in the two genetic models, or the activation of mTORC2 in different brain regions. For the EEG recordings, there are several other features that could be explored, like interictal spikes, seizure duration, or power spectra. Although these additions may not reveal a mechanism, they would give a much more complete picture.

Data on how Rictor loss affects basic properties of the mice is also needed. Prior studies showed Rictor loss shrinks neurons and dendrites, reduces the brain and body size, and causes motor impairments (PMID: 23569215). These variables could affect seizure detection and are important to evaluate Rictor as a viable drug target. They should be reported in these two models of Rictor loss.

The observation that mTORC2 may phosphorylate a sodium channel is interesting but would need to be further explored if it were to give evidence that it underlies the effect of Rictor loss on seizures. Prior evidence shows that phosphorylation of sodium channels is associated with a decrease in their activity. In this context mTORC2 inhibition would be expected to increase Na⁺ channel activity and exacerbate epilepsy.

It is likely that both mTORC1 and mTORC2 contribute to seizures and epilepsy. Many genetic causes of epilepsy, like TSC1/2 or DEPDC5, hyperactivate mTORC1, but inhibit mTORC2. Rapamycin has shown to be beneficial in these models, suggesting it also works by inhibiting mTORC1. A recent paper from the Bateup lab showed that Raptor, but not Rictor, loss

improves phenotypes of TSC1 loss. The abstract and the discussion of this paper don't present a balanced reading of the literature in this regard and should be rewritten as such.

The methods are very sparse and make it difficult to evaluate the paper without going back and reading several other papers. Also, the identification of seizures is of major importance to this paper but the methods merely state that "All EEG data including seizure frequency was interpreted by analysts with expertise in epilepsy who were blinded to genotype." There should be more objective/describable criteria to what is and isn't called a seizure.

I'm not sure what the difference is between the Western analysis in Figure 1 versus Supplementary Figure 1. That should be made more clear.

Reviewer #2 (Remarks to the Author):

A large proportion of epileptic patients do not respond to currently available interventions. The authors here investigate mTORC2 as a therapeutic target, building on a prior publication (Chen et al 2019), demonstrating a reduction in seizures and behavioral abnormalities in Pten deficient mice after mTORC2 activity reduction. Here, the authors demonstrate that genetic and antisense oligonucleotide inhibition of mTORC2 activity dramatically reduce the severity of seizures across two chemoconvulsant models and two genetic models of mTORopathy/epilepsy. Through mass spectrometry and immunoprecipitation experiments, the authors also identify a novel target of mTORC2, Nav1.2, which has a known mechanistic role in seizures, but the authors fail to link Nav1.2 to the rescue of seizure phenotype in their treatments. While these findings, overall, are very meaningful to understanding the mechanisms underlying seizures and a potential therapeutic target, this along with some other conceptual and design questions and concerns limit the impact of this study.

1. The results in figure 1 appear to be underpowered (n=4/ group). While it is apparent that mTORC2 activity is increased at the measured timepoints, the conclusion that mTORC1 is not increased at later timepoints is not valid without a sufficient sample size.
2. The authors do not state which figure (Fig 1 or S1) is from hippocampus and which is from cortex.

3. It is not clear what 0h refers to in figures 1 and S1. Is this directly after KA administration or in a naïve animal?
4. Does Rictor knockout in forebrain neurons produce an overt abnormal phenotype?
5. Does genetic or ASO reduction of mTORC2 activity affect mTORC1 activity or levels? If so, this would have significant implications for the long-term consequences of mTORC2 reduction and, thus, should be examined.
6. It is not stated when mice were injected with ASO for experiments in figure 3.
7. It is not clear when Akt was examined after PTZ administration.
8. mTORC1 activity should also be examined in PTZ experiments.
9. In addition to phospho-Akt/Akt and phospho-pS6/S6, the total levels of these proteins should also be presented (normalized to housekeeping protein).
10. Results in figure 5 suggest that mTORC2 phosphorylates Nav1.2, but there is no suggestion or discussion as to how this might affect neuronal excitability/seizure phenotype.
11. Further, the impact of this study would be substantially increased if results in figure 5 were expanded upon to determine whether there are changes in Nav1.2 phosphorylation and functional consequences of this alteration in neurons after reduction of mTORC2 activity in vivo.

Minor concerns

1. It is not stated how long mice in experiments of Figures 2 and 3 were EEG monitored.
2. In the last paragraph of the results section Rictor and Sin1 are discussed, but it is not clear which (or both) harbor a substrate binding site. This would be further clarified with a reference demonstrating this statement.
3. It appears that western blot data were normalized to 0h or controls, but this is never stated.
4. Some of the figures in the main PDF were cut off (figures 1, 3, and 4).

Reviewer #3 (Remarks to the Author):

The authors studied the role of mTORC2 in various animal models for epilepsy. They found that genetic deletion of mTORC2 from forebrain neurons was highly protective against kainic acid-induced behavioral and EEG seizures. Furthermore, they showed that selective

inhibition of mTORC2 with a specific antisense oligonucleotide (ASO) robustly suppressed spontaneous recurrent seizures in both pharmacological and genetic mouse models of epilepsy. Finally, using unbiased phosphoproteomics analyses and immunoprecipitation experiments, they identified and validated a new selective target of mTORC2, Nav1.2, which has been implicated in epilepsy and neuronal excitability.

This is an interesting and well written manuscript, with novel data, elegant methodologies, and well performed experiments. Strong points are the use of various acute and chronic animal models of epilepsy, the ASO treatment and the unbiased phosphoproteomics analysis, which convincingly show that mTORC2 plays an important role in epileptogenesis and revealed a potential target, Nav1.2.

Specific remarks:

Introduction

The authors provide a description for epilepsy, which focusses on the occurrence of seizures. This is correct, however, too limited. Epilepsy includes more than that, e.g., the neurobiologic, cognitive, psychological, and social consequences of the disease (see ILAE definition of epilepsy, Fisher 2005). Please add these elements.

Please replace “antiepileptic drugs” with “anti-seizure medication”.

A few studies are cited in which rapamycin (or related compounds) was used in genetic models of mTORopathies and models of TLE. However, there are many more studies, which are nicely summarized by Citraro et al. (Pharm Res 2016). By including this review, a better overview of the field is provided.

Methods

Did (some of) the KA injected mice develop status epilepticus? If so, indicate this in the manuscript and describe the severity and duration.

Please, indicate more details about the EEG recordings (e.g., filter settings, sampling rate) and provide a definition for an electrographical seizure (e.g., change in amplitude/frequency as well as a minimal duration of a seizure).

We would like to thank the reviewers for the thoughtful and constructive criticisms of our manuscript. Point-by-point answers to the reviewers' comments are listed below.

Reviewer #1

1. **There is no evidence that mTORC2 is driving or playing a role in the development of seizures in these models. Other approaches to inhibit brain activity, like treating with barbiturates, would also likely suppress these seizures, but I wouldn't interpret that data as suggesting that GABA receptor dysfunction is causing seizures in these models. The paper should not make claims that mTORC2 activation causes seizures or plays a role in their development. The finding that Rictor loss lessens the severity of the acute KA seizures suggests that mTORC2 inhibition is more a general suppressor of brain excitability than a participant in epileptogenesis.**

We agree with the reviewer. We used the term “drive” to exemplify that by suppressing mTORC2 activity, we can control (reduce) seizure outcome in multiple models. That said, we apologize for any confusion. To reflect our findings more accurately throughout the manuscript, we have now changed the title to **“Targeted inhibition of mTORC2 suppresses seizures across diverse models of epilepsy”**.

2. **It is well known that mTORC2 activates mTORC1 via Akt-Tsc signaling. It is therefore possible that the protective effect of Rictor downregulation occurs via inhibition of mTORC1. This is not explored experimentally or discussed.**

We thank the reviewer for raising this point, for which we now include a more comprehensive discussion. Based on our previous and newly added data, we believe it is very unlikely that the seizure protective effects of mTORC2 inhibition are due to mTORC1 inhibition.

First, mTORC1 activity is not altered in the brain of *Rictor* fb-KO mice when *Rictor* has been deleted in the forebrain postnatally (**See new Supplementary Fig. 3**, Huang *et al.*, *Nat Neurosci.* 2013) or in neurons during development (Thomanetz *et al.*, *JCB.* 2013).

Second, and in agreement with the genetic findings, Rictor-ASO treatment reduced mTORC2 (but not mTORC1) activity in the brain (**See new Supplementary Fig. 4**).

Third, our results are consistent with the widely accepted concept that inhibition of mTORC2 has no effect on mTORC1 activity across different cell types (Guertin *et al.*, *Dev Cell.* 2006; Shiota *et al.*, *Dev Cell.* 2006; Jacinto *et al.*, *Cell* 2006; Carson *et al.*, *Hum Mol Gen.* 2013; Thomanetz *et al.*, *JCB.* 2013) and organisms (Yang *et al.*, *Genes & Dev.* 2006). That said, we acknowledge that virus-mediated deletion of *Rictor* in cultured neurons reduces mTORC1 activity (Karalis *et al.*, *Nat Comm.* 2022; McCabe *et al.*, *eLife* 2020). Whether this is a cell type or context specific effect remains to be determined.

Finally, and more importantly, our preliminary results show that genetic inhibition of mTORC1 in forebrain neurons (*Rptor* fb-KO mice) did not protect against KA-induced seizures (**Appendix #1**). Indeed, our pilot experiments show an increased susceptibility to KA-induced behavioral seizures in *Rptor* fb-KO mice, which we will study in a separate manuscript. In conclusion, our results support the notion that the seizure suppression mediated by inhibition of mTORC2 is independent of mTORC1.

Appendix #1. Genetic inhibition of mTORC1 fails to protect against KA-induced behavioral seizures. Forebrain specific deletion of *Rptor* (*Rptor* fb-KO) reduces Raptor protein levels (**b**) and suppresses mTORC1 activity, as measured

by phosphorylation of S6 at Ser 240/244 (c). Additionally, our preliminary results show that mTORC1-deficient mice exhibit increased Racine score at 25mg/kg KA (d) and require a lower dose to exhibit stage 4 behavioral seizures.

3. **There is a lack of depth in the reporting of the results that seems like a missed opportunity to achieve further insight. With the use of Western blots, it shouldn't be too difficult to examine other targets of mTORC1 and mTORC2, the timeline or presence of mTORC2 activation in the two genetic models, or the activation of mTORC2 in different brain regions. For the EEG recordings, there are several other features that could be explored, like interictal spikes, seizure duration, or power spectra. Although these additions may not reveal a mechanism, they would give a much more complete picture.**

We thank the reviewer for this important suggestion. As requested, we have now included new data regarding the effects of genetic and ASO-mediated suppression of Rictor on the activity of mTORC1 and mTORC2 (See new Supplementary Fig. 3-4). mTORC1 activity was determined by measuring p-S6^{S240/244}, whereas mTORC2 activity by examining the activity of its three major downstream targets (p-Akt^{S473}, p-PKC^{S657} and p-NDRG1^{T346}). As expected, genetic (*Rictor* fb-KO mice, see new Supplementary Fig. 3) or ASO-mediated suppression of Rictor in the brain (See new Supplementary Fig. 4) reduced mTORC2 activity, but had no effect on mTORC1 activity. As indicated above, these results are consistent with previous findings that inhibition of mTORC2 activity has no effect on mTORC1 activity (Guertin *et al.*, *Dev Cell*. 2006; Shiota *et al.*, *Dev Cell*. 2006; Jacinto *et al.*, *Cell* 2006; Sarbassov *et al.*, *Science* 2005; Carson *et al.*, *Hum Mol Gen.* 2013; Thomantez *et al.*, *JCB*. 2013; Lamming *et al.*, *Aging Cell* 2014; Chen *et al.*, *Nat Med*. 2019).

In addition, as suggested by the reviewer, we have now measured the activity of the mTOR complexes in all the seizure models studied here. We found that while PTZ induced both mTORC1 and mTORC2 activity (See new Supplementary Fig. 5a-c), deletion of *Kcna1* failed to modulate mTOR complexes activity (See new Supplementary Fig. 6a-c). Furthermore, in the humanized mTOR model, only mTORC1 activity was increased (See new Supplementary Fig. 6f-h). Remarkably, Rictor-ASO reduced spontaneous seizures in all these models.

We agree with the reviewer that interictal spike numbers, seizure duration and interictal power spectra information might give a more complete picture. At the same time, the mechanistic relevance of these parameters to the number of ictal events remains unclear, which is why we focused on the actual ictal events themselves. For instance, interictal spike frequency has both pro-seizure and anti-seizure actions (Avoli *et al.*, *Epilepsy Curr*. 2006; Karoly *et al.*, *Brain* 2016) and does not predict seizure duration (Asadollahi *et al.*, *Clin EEG Neurosci* .2019) or seizure freedom (Lee *et al.*, *J Epilepsy Res*. 2019). Likewise, differences in power spectral bands are descriptive and depend as much upon behavioral state which is interesting but not a goal of this study.

4. **Data on how Rictor loss affects basic properties of the mice is also needed. Prior studies showed Rictor loss shrinks neurons and dendrites, reduces the brain and body size, and causes motor impairments (PMID: 23569215). These variables could affect seizure detection and are important to evaluate Rictor as a viable drug target. They should be reported in these two models of Rictor loss.**

It is noteworthy that in the paper cited by the Reviewer (PMID: 23569215), *Rictor* has been removed in neurons during development, which is likely the reason why it has a significant effect on brain and body size. In contrast, when we removed *Rictor* from forebrain neurons postnatally (*Rictor* fb-KO mice), we found that there were no gross brain abnormalities or changes in the expression of several synaptic markers in *Rictor* fb-KO mice (see Huang *et al.*, *Nat Neurosci*. 2013, see Appendix #2). In addition, we found that basic synaptic properties are normal in *Rictor* fb-KO mice (see Huang *et al.*, *Nat Neurosci*. 2013, see Appendix #2).

[figure redacted]

Appendix #2. Genetic inhibition of mTORC2 led to no gross brain abnormalities or changes in synaptic markers and basic synaptic transmission (From Supplemental Figures 2 and 4 of Huang *et al.*, *Nat Neurosci.* 2013). (a) Nissl stain from control (WT) and *Rictor* fb-KO shows no gross brain abnormality and no changes in synaptic markers (b-c). *Rictor* fb-KO mice also showed normal basic synaptic transmission compared to control (WT) mice as quantified by afferent volley in response to stimulation, (d) fEPSP slope for afferent volleys (e) and paired pulse ratio (f).

Additionally, we have also found that Rictor-ASO is safe and tolerable in mice (Chen *et al.*, *Nat Med.* 2019, **Appendix #3**). As requested by the reviewer, we have now reported these findings in the manuscript (**See discussion, page 11**).

[figure redacted]

Appendix #3. ASO-mediated inhibition of mTORC2 does not affect weight or increase gliosis or inflammatory markers (From Extended Data Fig 9. of Chen *et al.*, *Nat Med.* 2019). Rictor-ASO treated mice (a) showed no change in weight gain several weeks after injection (b) and no increase in markers of gliosis (GFAP) or inflammation (Aif1, CD68) (c).

5. **The observation that mTORC2 may phosphorylate a sodium channel is interesting but would need to be further explored if it were to give evidence that it underlies the effect of Rictor loss on seizures. Prior evidence shows that phosphorylation of sodium channels is associated with a decrease in their activity. In this context mTORC2 inhibition would be expected to increase Na⁺ channel activity and exacerbate epilepsy.**

We understand the concern over this point. It is first important to note that both gain-of-function and loss-of-function mutations in Nav1.2 are linked to seizures (Brunklaus *et al.*, *Epilepsia* 2020; Miao *et al.*, *Front Neurol.* 2021; Reynolds *et al.*, *Eur J Paediatr Neurol.* 2020), and in addition, here, we have identified 2 “new” phosphorylation sites that have not previously been functionally studied.

We believe that addressing whether mTORC2-mediated Nav1.2 phosphorylation underlies the effects of Rictor loss is beyond the scope of this paper since a) we will have to generate new knock-in mice carrying mutations in Nav1.2 where the newly identified serine phosphosites will have to be mutated to alanine and b) we will have generate phospho-specific antibodies against those sites, which will take several years to achieve. Moreover, the knock-in mice would have to be crossed with mTORC2-deficient mice to assess whether the effect of mTORC2 is only due to Nav1.2. During a call/discussion with the editor of this journal regarding the revision, she agreed that this issue does not need to be addressed at this time.

That said, here we show selective binding of Nav1.2 to the mTORC2-specific adaptor proteins Rictor and Sin1 and therefore simply propose this voltage-gated channel as a target of mTORC2 in the brain, rather than a singular explanation of the seizure mechanism. Finally, as we indicated in the discussion (**See discussion, pages 10-11**), we hypothesize that mTORC2 could regulate neuronal rhythmicity by phosphorylating more than one downstream target.

6. **It is likely that both mTORC1 and mTORC2 contribute to seizures and epilepsy. Many genetic causes of epilepsy, like TSC1/2 or DEPDC5, hyperactivate mTORC1, but inhibit mTORC2. Rapamycin has shown to be beneficial in these models, suggesting it also works by inhibiting mTORC1. A recent paper from the Bateup lab showed that Raptor, but not Rictor, loss improves phenotypes of TSC1 loss. The abstract and the discussion of this paper don't present a balanced reading of the literature in this regard and should be rewritten as such.**

We agree with the reviewer that the identity and function of the upstream mTORopathy-causing mutation may determine the relevance of mTORC1 or mTORC2. For instance, in *Tsc1/2*, *DEPDC5* or *STRADA* related seizure disorders, mTORC1 may be the major driver since mTORC1 is hyperactive and mTORC2 activity is often reduced or unchanged. In contrast, in *Pten*-related epilepsy, we found that inhibition of mTORC2, but not mTORC1, reduced seizures (Chen *et al.*, *Nat Med.* 2019).

As per the very interesting paper referenced by the reviewer (Karalis *et al.*, *Nat Comm.* 2022), the authors show that deletion of *Tsc1* led to hyperactivation of mTORC1 and inhibition of mTORC2, as expected (Zhang *et al.*, *JCI.* 2003). Interestingly, the partial deletion of *Rptor* rescued the *Tsc1*-related phenotypes normalizes both mTORC1 and mTORC2 activity in *Tsc1*-deficient mice. The decrease in mTORC1 activity is due to the absence of the adaptor protein Raptor and the increase in mTORC2 activity upon *Rptor* deletion is due to the absence of the inhibitory feedback onto IRS1-PI3K, which is an inherent feature of mTOR signaling (Polak *et al.*, *Cell Metab.* 2008; Bentzinger *et al.*, *Cell Metab.* 2008; Shende *et al.*, *Circulation* 2011; Manning *JCB.* 2004). Thus, given that in this model, deletion of *Rptor* rebalances both mTORC1 and mTORC2 activity, it remains unclear whether the “rescued” phenotype is due to normalization of either mTORC1 or mTORC2 levels. Moreover, in this study, deletion of *Rictor* failed to rescue the *Tsc1*-related phenotypes, likely because mTORC2 activity was already reduced in *Tsc1* lacking cells. To rule out mTORC2 as a major player in *Tsc1*-mediated pathology, one should selectively increase (not reduce) mTORC2 activity, which we understand is technically challenging.

As requested by the reviewer, we have now added this paper to the discussion and include a paragraph where we postulate the new idea that seizure disorders may be stratified based on their mTORC1 or mTORC2 dependency (**See discussion, page 9**).

In conclusion, our results that *Rictor* fb-KO mice and Rictor-ASO treated mice exhibit unchanged mTORC1 activity support the notion that the antiseizure effects of genetic or ASO-mediated inhibition of mTORC2 are independent of inhibition of mTORC1. Finally, we believe that our work can help to better define mTORC1 and mTORC2-related epilepsies.

7. **The methods are very sparse and make it difficult to evaluate the paper without going back and reading several other papers. Also, the identification of seizures is of major importance to this paper but the methods merely state that “All EEG data including seizure frequency was interpreted by analysts with expertise in epilepsy who were blinded to genotype.” There should be more objective/describable criteria to what is and isn't called a seizure.**

We thank the reviewer for raising this point. As suggested, we have now added a more detailed description on the methods including the criteria for seizure analysis and classification (**See methods**).

8. **I'm not sure what the difference is between the Western analysis in Figure 1 versus Supplementary Figure 1. That should be made more clear.**

Figure 1 refers to data from the cortex while Supplementary Figure 1 refers to data from the hippocampus. We apologize for the confusion and have rectified this in the manuscript.

- 1. The results in figure 1 appear to be underpowered (n=4/ group). While it is apparent that mTORC2 activity is increased at the measured timepoints, the conclusion that mTORC1 is not increased at later timepoints is not valid without a sufficient sample size.**

As suggested, we have now added more biological replicates and the results support our original findings (See new Fig. 1 and Supplementary Fig. 1).

- 2. The authors do not state which figure (Fig 1 or S1) is from hippocampus and which is from cortex.**

We apologize for this oversight. We now specify that Fig. 1 refers to data from the cortex while Supplementary Fig.1 refers to data from the hippocampus.

- 3. It is not clear what 0h refers to in figures 1 and S1. Is this directly after KA administration or in a naïve animal?**

Time 0h refers to naïve mice. We apologize for the confusion and have now clarified this point in the manuscript.

- 4. Does Rictor knockout in forebrain neurons produce an overt abnormal phenotype?**

We have previously showed that postnatal forebrain-specific deletion of *Rictor* did not cause gross brain abnormalities or basic changes in synaptic properties (Appendix #2). However, *Rictor* fb-KO mice exhibit deficits in long-term memory (Huang *et al.*, *Nat Neurosci.* 2013). Interestingly, we found that Rictor-ASO-mediated inhibition of mTORC2 in the adult brain was safe (Chen *et al.*, *Nat Med.* 2019, see Appendix #3) and had no effect on either motor activity levels or learning and memory (See new Supplementary Fig. 7). Taken together, our data shows that Rictor-ASO treatment effectively and safely reduced mTORC2 activity and suppressed seizures.

- 5. Does genetic or ASO reduction of mTORC2 activity affect mTORC1 activity or levels? If so, this would have significant implications for the long-term consequences of mTORC2 reduction and, thus, should be examined.**

We thank the reviewer for their comments and have now added data regarding the effects of genetic or ASO-mediated suppression of Rictor on mTORC1 and mTORC2 activity (See new Supplementary Figs. 3-4).

- 6. It is not stated when mice were injected with ASO for experiments in figure 3.**

ASO was injected at 6-weeks of age. (See new Supplementary Fig. 4).

- 7. It is not clear when Akt was examined after PTZ administration.**

We measured mTORC1 and mTORC2 activity 30 minutes after PTZ injection. (See methods).

- 8. mTORC1 activity should also be examined in PTZ experiments.**

As requested, we have now measured mTORC1 activity in the PTZ model (See Supplementary Fig. 5).

- 9. In addition to phospho-Akt/Akt and phospho-pS6/S6, the total levels of these proteins should also be presented (normalized to housekeeping protein).**

We respectfully disagree with the reviewer on this issue. According to molecular biology standards, the quantification of phosphoproteins using western blot should be done by measuring the ratio of phosphorylated vs. total protein (e.g., Talos *et al.*, *Ann Neurol.* 2018; Nguyen *et al.*, *Epilepsia* 2015; Zeng *et al.*, *J Neurosci.* 2009; Sarbassov *et al.*, *Mol Cell.* 2006; Chen *et al.*, *Nat Med.* 2019; Klofas *et al.*, *Hum Mol Gen.* 2020; Zhu *et al.*, *Nat Neurosci.* 2018; Zhu *et al.*, *Cell* 2011; Huang *et al.*, *Nat Neurosci.* 2013; Di Prisco *et al.*, *Nat Neurosci.* 2014 ; Zhu *et al.*, *Science* 2019) on the same membrane as we did here (please appreciate the same migration pattern for both phosphorylated and total proteins since our western blots were run on the same membrane).

- 10. Results in figure 5 suggest that mTORC2 phosphorylates Nav1.2, but there is no suggestion or discussion as to how this might affect neuronal excitability/seizure phenotype.**

We thank the reviewer for raising this point. We have now included a section in the discussion expanding on the possible mechanism by which mTORC2 may regulate Nav1.2 (See discussion, pages 10-11).

11. Further, the impact of this study would be substantially increased if results in figure 5 were expanded upon to determine whether there are changes in Nav1.2 phosphorylation and functional consequences of this alteration in neurons after reduction of mTORC2 activity in vivo.

As requested, we have added a new section in the discussion expanding on the possible roles of the phosphorylation of Nav1.2 by mTORC2 (**See discussion, pages 10-11**). Unfortunately, as there are no phospho-specific antibodies available, we are not able to validate the identified phosphorylation events *in vivo*.

Minor concerns

1. It is not stated how long mice in experiments of Figures 2 and 3 were EEG monitored.

As requested, we have now specified that EEG recording was done for 4hrs for the KA experiments (**See methods**).

2. In the last paragraph of the results section Rictor and Sin1 are discussed, but it is not clear which (or both) harbor a substrate binding site. This would be further clarified with a reference demonstrating this statement.

We have now added that Sin1 is proposed to harbor the substrate binding site (**See discussion**).

3. It appears that western blot data were normalized to 0h or controls, but this is never stated.

The data were normalized to untreated (ctrl) mice. (**See methods and figure legends**).

4. Some of the figures in the main PDF were cut off (figures 1, 3, and 4).

We apologize for this and have addressed it in the new manuscript.

Reviewer #3

1. Introduction:

The authors provide a description for epilepsy, which focusses on the occurrence of seizures. This is correct, however, too limited. Epilepsy includes more than that, e.g., the neurobiologic, cognitive, psychological, and social consequences of the disease (see ILAE definition of epilepsy, Fisher 2005). Please add these elements.

As suggested by the reviewer, we have added an updated definition of epilepsy in the introduction (**See introduction, page 3**).

Please replace “antiepileptic drugs” with “anti-seizure medication”.

We have now corrected this phrase as suggested by the reviewer.

A few studies are cited in which rapamycin (or related compounds) was used in genetic models of mTORopathies and models of TLE. However, there are many more studies, which are nicely summarized by Citraro et al. (Pharm Res 2016). By including this review, a better overview of the field is provided.

As requested, we now include the suggested review article.

2. Methods:

Did (some of) the KA injected mice develop status epilepticus? If so, indicate this in the manuscript and describe the severity and duration.

While we did not measure status epilepticus in our paradigm, we quantified the survival post KA-injection and found that mTORC2-deficient mice (*Rictor* fb-KO and *Rictor*-ASO treated mice) exhibited reduced mortality following the KA-challenge compared to control mice (**Appendix #4**).

Please, indicate more details about the EEG recordings (e.g., filter settings, sampling rate) and provide a definition for an electrographical seizure (e.g., change in amplitude/frequency as well as a minimal duration of a seizure).

As suggested by the reviewer, we have now added a more detailed description for EEG recording and seizure definition (**See methods**).

REVIEWERS' COMMENTS

Reviewer #2 (Remarks to the Author):

In the revised manuscript, the authors addressed the majority of questions raised. A couple of concerns remain which are listed below (based on previous comments):

1. No further comments.
2. No further comments.
3. No further comments.
4. The behavioral experiments (Fig S7) were never referenced in the results of the paper.
5. No further comments.
6. No further comments.
7. These experiments were performed 30 minutes after PTZ administration. This timepoint may miss some of the important sequelae of events that take place in the hours/days following seizure kindling. The evolution of changes would be important in fully understanding the longer term impact. Timepoints that mirror the KA experiments would allow for the determination of whether mTORC1/mTORC2 activity levels are similar across chemoconvulsant models.
8. No further comments.
9. No further comments.
10. No further comments
11. The authors did not explore functional consequences of Nav1.2 phosphorylation, which would add substantial impact to the paper, but explain that this may be out of the scope of study. Nevertheless, this leaves a critical question regarding mechanism unresolved.

Minor concerns

1. No further comments.
2. No further comments
3. No further comments
4. No further comments.

Reviewer #3 (Remarks to the Author):

The authors satisfactorily answered all my questions and revised the manuscript accordingly